# Beyond Valence and Arousal: The Role of Age of Acquisition in Emotion Word Recognition

**DOI:** 10.3390/bs13070568

**Published:** 2023-07-09

**Authors:** Chenggang Wu, Yiwen Shi, Juan Zhang

**Affiliations:** 1Key Laboratory of Multilingual Education with AI, School of Education, Shanghai International Studies University, Shanghai 200083, China; chenggangwu@shisu.edu.cn (C.W.); 0181114004@shisu.edu.cn (Y.S.); 2Institute of Linguistics, Shanghai International Studies University, Shanghai 200083, China; 3Faculty of Education, University of Macau, Macau, China; 4Centre for Cognitive and Brain Sciences, University of Macau, Macau, China

**Keywords:** valence, arousal, emotion word, age of acquisition

## Abstract

Although the age of acquisition (AoA) effect has been established in numerous studies, how emotion word processing is modulated by AoA, along with affective factors, such as valence and arousal, is not well understood. Hence, the influence of age of acquisition (AoA), valence, and arousal on Chinese emotion word recognition was investigated through two experiments. Experiment 1 (*N* = 30) adopted a valence judgment task to explore the roles of valence and AoA in emotion word recognition, whereas Experiment 2 (*N* = 30) used a lexical decision task to examine AoA and arousal effects. A mixed linear effects model was used to examine the fixed effects of AoA, arousal, and valence and random effects of participants and items. The findings provided confirmation of the effects of AoA, valence, and arousal. Notably, AoA and valence had independent influences on emotion word recognition, as evidenced by longer reaction times for later-acquired words and negative words compared to early-acquired words and positive words (all *ps* < 0.05). On the other hand, AoA and arousal demonstrated interdependent effects on emotion word recognition. Specifically, a larger AoA effect was observed for low-arousing words (all *ps* < 0.05), whereas the influence of AoA on high-arousing words was insignificant. These results underscored the significance of AoA in processing emotion words and highlighted the interplay between AoA and arousal. Additionally, it is plausible to suggest that the AoA effect was primarily perceptual rather than semantic in nature.

## 1. Introduction

It has long been recognized that emotions can be characterized by a two-dimensional construct, encompassing valence and arousal [1,2]. Valence categorizes emotion words into positive (e.g., joy) and negative (e.g., fear), while arousal measures the intensity of the effect conveyed by these words (e.g., *parturition* for a high-arousing word, *windmill* for a low-arousing word). Within this framework, research on emotion word recognition has primarily focused on exploring the interactive relationship between valence and arousal [3,4]. Emotion words, by definition in the two dimensions, are more positive or negative than neutral words, and also are more arousing than neutral words. However, recent studies have revealed that, apart from valence and arousal, there are several other relevant variables that can modulate emotion word processing, including concreteness [5], age of acquisition [6], emotion prototypicality [7], and emotion word type [8,9,10,11,12,13,14,15,16]. Concreteness differentiates concrete words (e.g., table) from abstract words (e.g., destiny). Age of acquisition refers to the year of onset of learning a word. For example, the word “tree” is learned earlier than the word “parturition”. Emotion word type divides emotion-label words (e.g., sadness) and emotion-laden words (e.g., death). Despite the growing body of research in this area, the investigation of age of acquisition (AoA) in relation to emotion word processing has been limited, hindering our understanding of the developmental interplay between emotion and language. Compared to the newly discovered factors influencing emotion word recognition, such as concreteness, emotional prototypicality, and emotion word type, valence and arousal have received considerable attention in previous research and have been extensively studied. Based on the robust findings of the effects of valence and arousal, this study aims to examine how AoA, in conjunction with valence and arousal, influences the recognition of emotion words.

### 1.1. Valence, Arousal, and Beyond: Collecting Factors Influencing Emotion Words

Hedonic valence, which represents the pleasantness or unpleasantness of emotion words, plays a crucial role in emotion word processing. Numerous studies have focused on investigating the influence of valence on emotion word recognition, consistently revealing a positive bias towards positive words. This bias is evident in tasks such as lexical decision and semantic categorization, where positive words are processed more rapidly and accurately compared to negative words [17]. For instance, Kazanas and Altarriba [18] found that positive words were recognized faster than negative words in the lexical decision task. This advantage for positive words has been replicated in subsequent studies [18,19,20]. Similarly, in semantic categorization tasks, where participants retrieve semantic information from words, approximately half of the reviewed studies supported the positive bias. For example, Zhang et al. [12] reported that Chinese negative words had longer reaction times than positive words in a valence judgment task, a finding that was replicated in Polish–English and Romanian–English bilinguals [21]. However, it is important to note that contradictory evidence exists, with some studies showing no difference between positive and negative words or even a negative bias, where negative words are processed faster than positive words [17]. For example, Kousta et al. [22] observed that positive and negative words both exhibited similar processing advantages over neutral words in lexical decision tasks. This suggests that previous studies demonstrating symmetry between positive and negative words may have failed to adequately control essential variables that influence emotion word processing [22]. Additionally, it is worth noting that Kousta et al. [22] included neutral words in their experiments, whereas Bromberek-Dyzman et al. [21] and Zhang et al. [12] did not include a neutral word condition. The inclusion or exclusion of neutral words in the stimulus construction could potentially influence the impact of valence on emotion word recognition. The selection of stimuli involves not only the presence or absence of specific stimuli but also the careful control of other variables. One such variable that can interact with valence and jointly impact emotion word processing is arousal. Larsen et al. [23] demonstrated that not all negative words were processed slower than positive words; rather, the presence of arousal modulated the speed of lexical decision. Specifically, negative words with moderate to low arousal were processed more slowly than positive words, whereas high-arousing negative words showed only nuanced differences compared to positive words [23]. These findings emphasize the significance of arousal as an important factor in emotion word recognition. For example, Bayer et al. [24] found that high-arousing words elicited larger electrophysiological activation than low-arousing words, suggesting that arousal affects both behavioral performance and brain responses [24].

In comparison to valence studies, research on arousal is relatively scarce, and previous findings on its impact have been mixed, necessitating further clarification. Some studies have found no significant influence of arousal on emotion word processing [25]. However, Kever et al. [26] recently reported that in a constructive word recognition task, where participants determine whether a target word is related to emotions, high-arousing words were recognized more quickly than low-arousing words [26], corroborating the results of Recio et al. [27], who showed that arousal facilitated emotion word recognition [27]. Conversely, Kuperman et al. [28] revealed that high-arousing words were recognized more slowly than low-arousing words [28], indicating an inhibitory effect of arousal on word processing. Furthermore, increasing research has highlighted the interaction between arousal and valence [29]. For instance, Bayer et al. [29] demonstrated that negative low-arousing words were recognized more slowly than negative high-arousing words and positive low-arousing words. However, no significant difference in processing speed was observed between low-arousing and high-arousing words for positive words, suggesting that arousal primarily affects negative words rather than positive words. Similarly, Robinson et al. [30] consistently observed an interaction between valence and arousal in seven experiments [30]. They used emotion words as stimuli and found that negative high-arousing words were evaluated faster than negative low-arousing words, while positive low-arousing words had higher processing speed than positive high-arousing words. Another study by Citron et al. [3] also demonstrated an interaction between valence and arousal. They reported that positive low-arousing and negative high-arousing words were recognized faster than positive high-arousing and negative low-arousing words [3]. These findings indicate that arousal facilitates the processing of negative words but inhibits the processing of positive words.

In addition to valence and arousal, recent studies have highlighted the influence of other factors on emotion word processing, including concreteness [5], emotional prototypicality [7], and emotion word type [31]. For example, Yao et al. [32] manipulated the valence and concreteness of emotion words in a lexical decision task and observed that emotion words were processed faster than neutral words, with a greater advantage for concrete words compared to abstract words [32]. They further investigated the role of word imageability and found that the interaction between emotion and concreteness was mediated by imageability. Specifically, the effect of emotion was significant for abstract words with high imageability but not for abstract words with low imageability. However, the emotion effect for concrete words was not related to imageability. These findings suggest that different aspects of concreteness and imageability play a role in modulating emotion effects during visual word recognition.

Despite the increasing number of studies exploring various factors influencing emotion word processing, one important factor has been overlooked: the age of acquisition (AoA). A recent review highlighted the lack of proper control of AoA in previous studies, making it challenging to determine the interplay between AoA and emotion word processing [6]. The following section will introduce the AoA effect and its association with emotion word processing.

### 1.2. AoA Saliently Shapes Word Recognition, but Why Does AoA Matter for Emotion Words?

Carroll and White [33] conducted an initial investigation on how the age of acquisition (AoA) influences word naming and found that words learned at an early age were named faster than those learned later. This study suggested that AoA, along with word frequency, was an important factor shaping word processing [33]. Subsequent studies have replicated this effect in picture naming tasks across different languages, including Chinese [34], French [35], Spanish [36], and Persian [37]. For instance, Bonin et al. (2004) conducted a multiple regression study on the AoA effect in French word reading and found that both rated AoA and objective AoA significantly predicted French word naming. Likewise, Liu et al. [34] observed that both rated and objective AoA could predict Chinese object naming. The AoA effect has also been confirmed in lexical decision tasks in English [38], Chinese [39], Spanish [40], and English [38]. Cortese et al. [38] recently found that AoA predicted performance in lexical decision and reading aloud tasks even after controlling for related semantic variables [38].

To explain the AoA effect, three accounts have been proposed, but it is noteworthy that none of them consider the role of emotion. The multiple loci account suggests that the AoA effect is not limited to a single linguistic structure (e.g., phonology) but arises from multiple levels, including phonology, semantics, and perception/orthography [41]. In one study, five experiments (picture verification task, real/chimeric classification task, word–picture verification task, naming task, and delayed naming task) were conducted to examine the multiple loci account and found evidence for at least two loci of AoA. The AoA effect was greater in the naming task compared to the picture verification task, indicating that the requirement of accessing phonological representation amplifies the AoA effect [42]. Another account proposed by Brysbaert and Ghyselinck [43] suggests that the AoA effect consists of two components: a frequency-dependent component and a frequency-independent component. The former operates at the phonological and orthographic input level, while the latter operates at the semantic and conceptual input level [43]. This account also considers learning sequence and semantic nodes, suggesting that early-acquired words have an advantage in terms of lateral inhibition and richer semantic representations compared to later-acquired words. The third account is the mapping theory, which posits that the AoA effect arises from connections between different levels of lexical representation [44]. In the early stages of word acquisition, the neural network is highly plastic, but as new words are acquired and incorporated into the network, its plasticity decreases. However, Monaghan and Ellis [44] demonstrated that later-acquired words, if they share similar orthography-to-phonology mapping with early-acquired words, can leverage the knowledge stored in the early-acquired words and thus show no difference from them. In other words, the AoA effect becomes more pronounced when the orthography-to-phonology mapping is arbitrary [45].

Although these three accounts have received substantial support and have guided research on the AoA effect for decades, none of them take into account the influence of emotions on the AoA effect. Recent studies on emotion development have highlighted the importance of language in the formation of emotion concepts [46]. Similarly, there is evidence indicating that emotions also play a role in the formation of abstract concepts [47]. These studies have convincingly demonstrated an interplay between language and emotion development. For instance, Ponari et al. [48] examined the relationship between AoA and concreteness to determine if abstract words acquired early also have emotional connotations. They collected data on children’s performance in a lexical decision task involving abstract and concrete words, and their findings indicated that valence facilitated the recognition of abstract words more than concrete words [48]. These results further support the notion of a close connection between AoA and emotion. However, these studies did not experimentally investigate how emotional variables such as valence and arousal interact with AoA in the context of visual word recognition. Here, the term “interaction” refers to the interdependent influence of two independent variables on the dependent variables. When there is no interaction between the two factors, it indicates that each independent variable independently influences the dependent variables.

### 1.3. The Present Study

Drawing on this summary of previous studies, it is evident that age of acquisition (AoA) plays a significant role in shaping word recognition. However, experimental research on emotion word processing has not adequately controlled for AoA. As argued by Hinosoja et al. [6], AoA should be controlled for in emotion word research. However, this argument has not been explicitly confirmed. In other words, the impact of AoA, along with valence and arousal, on emotion word processing remains unclear. The present study aims to explore how AoA and emotion-related variables (valence and arousal) interactively influence visual word recognition through two experiments. One experiment focuses on valence and AoA, while the other concentrates on arousal and AoA. These experiments also employ different tasks to access two levels of representation.

Experiment 1 employs a valence judgment task where participants are required to judge the valence (positive or negative) of target words. In this task, participants need to retrieve semantic information, thus associating the AoA effect with semantic processing. In contrast, Experiment 2 utilizes a lexical decision task where participants must retrieve lexical information (such as orthography and phonology) to determine whether the target word is a real word or not. The distinct focuses of the two tasks will shed light on the role of AoA at the conceptual level (Experiment 1) and perceptual representation level (Experiment 2). Additionally, the experiments examine the moderating roles of valence and arousal in the AoA effect simultaneously within the same group of Chinese native speakers. The order of the two experiments was counterbalanced.

## 2. Experiment 1

The present experiment aimed to explore the role of AoA and valence in emotion word recognition in a valence judgment task. It was predicted that positive words would be processed faster than negative words and early-acquired words would have a higher processing speed than later-acquired words. If the AoA effect and valence were both at the conceptual level in this task, the two factors would interactively influence word recognition. Specifically, according to the density hypothesis [49], positive words are thought to be represented in a dense manner, while negative words are represented in a more dispersed manner. As a result, the representation of early- and later-acquired words differs for positive and negative words. Due to the dense representation of positive words, both early- and later-acquired words are closely connected and share similar associations. In contrast, for negative words, early-acquired words benefit from network plasticity and possess stronger semantic representations compared to later-acquired words. This advantage is particularly pronounced due to the dispersed nature of negative word representations. Consequently, the influence of age of acquisition (AoA) is assumed to have a larger effect on negative words compared to positive words. However, if the AoA effect is at the perceptual level (form representation) while the valence is at the semantic level (because participants need to retrieve valence information to finish this task), the two factors will independently influence the target word recognition. In other words, the representation density difference for negative words and positive words is not related to AoA, leading to no influence from AoA on valence.

### 2.1. Method

#### 2.1.1. Participants

Thirty native Chinese speaker undergraduates (mean age: 18.87 ± 3.83 years, 9 male) finished the experiment. Three participants reported that they were left-handed. None of them suffered from neurological or mental disorders. All of the participants signed the consent forms before the experiment. All of the participants speak English as their second language (L2).

#### 2.1.2. Materials

A total of 160 two-character simplified Chinese emotion compound words were obtained from a recent Chinese normative AoA database [50]. We selected early-acquired words that have a learning onset at around 9 years old and later-acquired words that have a learning onset at 14 years old. The 160 Chinese words were divided into four groups (positive and early-acquired words, PE; negative and early-acquired words, NE; positive and later-acquired words, PL; negative and later-acquired words, NL, with each group having 40 words. The four groups of words were matched on arousal (valence: *F* (1, 39) = 0.055, *p* > 0.1; AoA: *F* (1, 39) = 0.674, *p* > 0.1; the interaction between valence and AoA: *F* (1, 39) = 0.440, *p* > 0.1), number of strokes (valence: *F* (1, 39) = 0.007, *p* > 0.1; AoA: *F* (1, 39) = 0.015, *p* > 0.1; the interaction between valence and AoA: F (1, 39) = 1.842, *p* > 0.1), and word frequency (valence: *F* (1, 39) = 0.737, *p* > 0.1; AoA: *F* (1, 39) = 1.328, *p* > 0.1; the interaction between valence and AoA: *F* (1, 39) = 0.087, *p* > 0.1). The arousal (range: 0 to 4, 0 denotes “very low arousal” and 4 denotes “very high arousal”) and valence (range: −3 to 3, −3 denotes “extremely negative” while 3 denotes “extremely positive) values were retrieved from a recent Chinese normative database [51] and word frequency was obtained from SUBTLEX-CH [52]. Table 1 shows the characteristics of critical words in Experiment 1. The valence of the critical words was determined by the fact that the mean of positive words was larger than 1.7 whereas the mean of negative words was smaller than −1.7 (see the word list in Appendix A).

#### 2.1.3. Procedure

Prior to the experiment, participants provided signed consent forms. The experiment was implemented via E-Prime 1.1 on a laptop in the Key Laboratory of Multilingual Education with AI. Each trial began with a fixation presented for 1000 ms, followed by the presentation of a target word. Participants were instructed to judge the valence of the target word (whether the target word was a positive word or a negative word). The maximum presentation time for each target word was set to 2000 ms. If participants failed to respond within this time frame, the target word would disappear. The sequence of trials was randomized, and the corresponding judgment keys were counterbalanced among participants. Participants were instructed to respond to the target words as quickly and accurately as possible. The whole experiment usually took 8 min to finish. Due to the short duration of the experiment, the trials were presented in one block. Between the two experiments, the participants could have a short break.

### 2.2. Results

Trials that exceeded M ± 2.5 SD (RT being larger than 1425 ms) were trimmed for subsequent analysis (92.92% data were kept for further analysis). A linear mixed effect model containing fixed effects (valence and AoA) and random effects (items and participants) was tested in JASP (version 0.16.4) for reaction times (see Table 2 for descriptive statistics). The valence effect was significant (*β* = 15.489, *SE* = 3.008, *t* = 5.15, *p* < 0.001) for negative words (743 ms) being processed slower than positive words (712 ms). However, for AoA (*β* = −6.936, *SE* = 3.737, *t* = −1.856, *p* = 0.073) and the interaction between the valence and AoA (*β* = 0.699, *SE* = 2.978, *t* = 0.235, *p* = 0.815), significance was not reached. For accuracy rate, because accuracy rate is binary data, a generalized linear mixed effect model was analyzed. Positive words (0.979) had a higher accuracy rate than negative words (0.958), *β* = −0.363, *SE* = 0.136, *t* = −2.678, *p* = 0.007. Early-acquired words (0.978) also had a higher accuracy rate than later-acquired words (0.961), *β* = 0.290, *SE* = 0.126, *t* = 2.294, *p* = 0.022. However, there was no interaction between the two factors, *β* = −0.205, *SE* = 0.124, *t* = −1.648, *p* = 0.099.

### 2.3. Discussion

Experiment 1 yielded robust effects of valence and AoA, demonstrating that early-acquired words were recognized more accurately than late-acquired words, and positive words were recognized faster and more accurately than negative words. These findings are consistent with previous studies utilizing the lexical decision task [18,53]. Importantly, even after controlling for word frequency, the AoA effect remained significant, suggesting that AoA and word frequency are relatively independent factors.

The processing advantage of positive words over negative words in the valence judgment task has been consistently observed in previous investigations [8,9]. Wu et al. (2021a, 2021b) borrowed the idea of the density hypothesis to explain this advantage, suggesting that positive words are represented more densely compared to negative words, which are represented with greater disparity [49].

In Experiment 1, there was no interaction between valence and AoA. This finding indicates that these two factors are unrelated, at least in the context of the valence judgment task, which involves semantic processing. In this task, individuals need to differentiate between positive and negative words, suggesting that the activation and elaboration of positive and negative semantics play a crucial role [12]. Therefore, the results are closely aligned with the representation of positive and negative concepts and the density hypothesis [54,55]. If the AoA effect were also at the conceptual level, one would expect an interaction between these factors. However, the findings contradict this hypothesis. The notion that the AoA effect operates at the perceptual level was examined in the next experiment.

## 3. Experiment 2

Experiment 2 focused on the effect of AoA and arousal. In this experiment, a lexical decision task was adopted to explore whether AoA and arousal jointly influenced emotion word recognition. Lexical decision tasks require less semantic processing than valence judgment tasks, and participants could use orthography and phonology, which are at the perceptual level, to differentiate real words and non-words. Therefore, we expected that AoA and arousal would interact with each other in the lexical decision task, because the two factors are both at the perceptual level.

### 3.1. Method

#### 3.1.1. Participants

The sample for the second experiment was the same as for Experiment 1. The order of the two experiments was counterbalanced among the participants.

#### 3.1.2. Materials

The 160 Chinese words in this experiment were different from those in Experiment 1. The set of words were divided into four groups (high-arousing and early AoA, HE; high-arousing and late AoA, HL; low-arousing and early AoA, LE; and low-arousing and late AoA, LL), with 40 in each group. The four groups of words were matched on valence (arousal: *F* (1, 39) = 0.005, *p* > 0.1; AoA: *F* (1, 39) = 0.794, *p* > 0.1; the interaction between arousal and AoA: *F* (1, 39) = 0.188, *p* > 0.1), number of strokes (arousal: *F* (1, 39) = 0.069, *p* > 0.1; AoA: *F* (1, 39) = 0.100, *p* > 0.1; the interaction between arousal and AoA: F (1, 39) = 2.083, *p* > 0.1), and word frequency (arousal: *F* (1, 39) = 1.108, *p* > 0.1; AoA: *F* (1, 39) = 0.775, *p* > 0.1; the interaction between arousal and AoA: *F* (1, 39) = 0.296, *p* > 0.1). The additional 160 Chinese pseudowords were created by randomly combining two Chinese characters (see Table 3 for characteristics of critical stimuli of Experiment 2).

#### 3.1.3. Procedure

The participants were instructed to decide whether the target words were real words or not. In this task, the participants need to process the target words based on their lexical knowledge. The participants needed to press the corresponding buttons for *yes* (real words) or *no* (non-words) responses. The rest of the procedure was the same as Experiment 1.

### 3.2. Results

Responses (3.21%) that exceeded 2.5 SD ± mean (RT > 1211 ms) were deleted for subsequent analysis. The linear mixed effect model containing fixed effects (arousal and AoA) and random effects (participants and items) was tested against reaction time data. The high-arousing words (649 ms) were processed faster than low-arousing words (687 ms), *β* = −19.092, *SE* = 4.234 *t* = −4.509, *p* < 0.001. The early-acquired words (655 ms) had a shorter reaction time than later-acquired words (680 ms), *β* = −12.359, *SE* = 4.959, *t* = −2.492, *p* = 0.016. There was also an interaction between arousal and AoA, *β* = 9.212, *SE* = 4.483, *t* = 2.055, *p* = 0.044. The post hoc comparisons showed that there was no difference between early- and later-acquired words for high-arousing words, *z* = −0.506, *p* > 0.1. However, for low-arousing words, the difference between early- and later-acquired words was significant, *z* = −3.028, *p* < 0.01 (for descriptive statistics for Experiment 2, see Table 4). The generalized mixed effect model containing fixed effects (arousal and AoA) and random effects (participants and items) was examined for accuracy rate data. The fixed effect of arousal was not significant, *β* = 0.113, *SE* = 0.224, *t* = 0.504, *p* > 0.1. However, early-acquired words (0.99) had a higher accuracy rate than later-acquired words (0.96), *β* = 0.528, *SE* = 0.222, *t* = 2.379, *p* < 0.05. The interaction between arousal and AoA was also significant, *β* = −0.488, *SE* = 0.207, *t* = −2.354, *p <* 0.05. Post hoc comparisons also showed that for high-arousing words, AoA did not influence word recognition, *z* = 0.156, *p* > 0.1, but for low-arousing words, AoA facilitated word recognition, *z* = −3.466, *p* < 0.01.

### 3.3. Discussion

In Experiment 2, we first confirmed the presence of both arousal and AoA effects, indicating that both factors facilitated word recognition. The existence of the AoA effect in the lexical decision task aligns with previous research findings [38,56]. Similarly, the observed arousal effect in Experiment 2 is consistent with prior investigations [27,57].

The novel contribution of this experiment is the identification of an interaction between arousal and AoA, revealing that the AoA effect is more prominent for low-arousing words compared to high-arousing words. This result suggests that both factors operate at the same representation level, allowing for mutual influence. High-arousing words are more salient at the perceptual level due to their heightened activation compared to low-arousing words. The greater activation of high-arousing words surpasses the impact of AoA on emotion word recognition. For high-arousing words, activation occurs rapidly and automatically, leading to effortless recognition without relying on early network establishment. However, for low-arousing words, the activation at the perceptual level is relatively lower. Consequently, early-acquired words demonstrate their superiority in terms of activation, benefiting from their superior representation at the perceptual level compared to late-acquired words. This advantage in terms of activation facilitates the recognition of low-arousing words. The interaction between arousal and AoA provides further support for the notion that the AoA effect operates at the perceptual level, indicating a potential modulation between AoA and arousal.

## 4. General Discussion

The aim of the present study was to investigate the interaction between AoA and valence (Experiment 1) and AoA and arousal (Experiment 2) in two different tasks: a valence judgment task (Experiment 1) and a lexical decision task (Experiment 2). Both experiments confirmed the effects of AoA, valence, and arousal on emotion word recognition. Importantly, AoA did not interact with valence in Experiment 1 but did interact with arousal in Experiment 2. These results suggest that in addition to valence and arousal, AoA also influences emotion word recognition in different tasks, and the extent of this influence may vary depending on the specific emotional factor.

### 4.1. AoA’s Impact on Emotion Word Recognition

It is widely acknowledged that early-acquired words are processed faster and more accurately than words acquired later in life [45]. However, the role of AoA in modulating emotion word processing remains largely unexplored. Hinojosa et al. [6] noted that many previous studies on emotion words did not adequately control for AoA [6]. In light of this observation, the present study investigated the influence of AoA on emotion recognition across two tasks and consistently found that emotion word recognition is influenced by AoA, with early-acquired emotion words being recognized more efficiently than later-acquired words. This result not only confirms the presence of the AoA effect but also extends its applicability from neutral words to emotion words, underscoring the importance of considering AoA in research on emotion word processing.

The relationship between AoA and emotion is likely interactive. Kousta et al. (2009) proposed the idea that emotions play a central role in the development of abstract word categories, as abstract words tend to be more emotionally charged than concrete words. Building on this notion, Ponari et al. [48] investigated the role of valence in the development of abstract words and found that both positive and negative abstract words are acquired earlier than neutral words, indicating the crucial role of emotion in the acquisition of abstract words. From a constructionist perspective on emotion, which posits that language shapes emotions and that children develop emotion categories through language learning, emotion development relies on language development [46,58]. These two perspectives highlight the interactive roles of emotion and language in development and align with the observed AoA effect on emotion word recognition. Emotion words acquired at an early stage benefit from a processing advantage over later-acquired words, reflecting the significant influence of learning sequences in language processing and representation. Early-acquired words exhibit greater neural plasticity within networks compared to later-acquired words, and this enhanced plasticity extends to emotion words as well. A recent normative study demonstrated a negative relationship between AoA and emotional prototypicality (EmoPro), which measures the extent to which an emotion word is prototypical. For example, the word “sadness” is a highly prototypical emotion word, while “death” is a less prototypical emotion word. This negative correlation suggests that early-acquired words are more prototypical emotion words compared to later-acquired words. Hence, the superiority of early-acquired emotion words over later-acquired ones can be attributed to EmoPro.

### 4.2. The Relationship between AoA and Emotional Factors

Another important finding of the present study was the interaction between AoA and arousal, whereas no interaction was observed between AoA and valence. Previous normative studies in different languages, such as Spanish, English, and Dutch, have explored the relationship between valence and AoA. For instance, Moors et al. [59] provided norms for the valence, arousal, and AoA of 4300 Dutch words and found a negative correlation between AoA and valence, but no correlation between AoA and arousal [59]. Similarly, Pérez et al. [60] reported a negative correlation between AoA and valence, but no association between AoA and arousal [60]. However, the present study revealed independent influences of AoA and valence on emotion word recognition. This discrepancy could be attributed to the word selection in the different studies. To ensure comparability across conditions, we specifically selected words with a mean AoA of 7–9 years for the early-acquired condition, while words with a mean AoA of 13–14 years were chosen for the later-acquired condition. Thus, the range of AoA in the present study was not as broad as that in the normative studies conducted in Spanish and Dutch. Additionally, our experimental design focused on investigating the roles of AoA and valence in the valence judgment task, whereas normative studies collected participants’ subjective ratings of AoA and valence. Therefore, the two tasks also differed in their task demands. A notable feature of the valence judgment task is its requirement for retrieving valence-related semantic information. The processing advantage of positive words over negative words has been consistently observed across various languages, including English [18] and Chinese [8]. If AoA and valence both operated at the semantic level and valence shaped emotion development [48], one would expect an interaction between the two factors. However, since no interaction was observed in the valence judgment task, it is plausible to suggest that the AoA effect operates at the perceptual level rather than the semantic level. This notion is further supported by the results of Experiment 2, which employed a lexical decision task. In such tasks, participants typically rely on retrieving perceptual-level information (phonology and orthography) to distinguish between words and non-words. Therefore, for words with high arousal, the facilitation of word form encoding due to early AoA is minimal [61], as high-arousing words already exhibit enhanced encoding and automatic recognition. Conversely, for low-arousing words, the encoding of word forms is weaker, allowing for the potential facilitation of early AoA in such encoding processes.

### 4.3. Limitations and Future Directions

The present study has several limitations that warrant further discussion. Firstly, the selection of words in our study was limited to early-acquired words at around 7–9 years and later-acquired words at around 13–14 years to ensure comparability across various dimensions. Therefore, caution should be exercised when interpreting the AoA effect in our study, as it is specific to the selected words. Future research could expand the investigation by including words with even earlier AoA to explore the association between AoA and affective factors more comprehensively.

Secondly, the neural mechanisms underlying the interplay between AoA and affective factors were not addressed in our study. Future studies could employ neuroimaging techniques to investigate how AoA and affective factors modulate emotion word recognition at the neural level. This would provide a deeper understanding of the underlying brain processes and further underscore the importance of considering AoA in emotion word research.

Finally, it is worth noting that affective factors extend beyond valence and arousal. One recent factor of interest is emotional prototypicality (EmoPro), which measures the extent to which a word represents a specific emotion [60]. Recent evidence has highlighted the role of EmoPro in enhancing emotion word processing [7]. Moreover, Pérez et al. [60] found a negative association between AoA and EmoPro, suggesting that earlier-acquired words are more prototypical of their corresponding emotions. However, how AoA and EmoPro jointly influence emotion word recognition remains unclear. Future studies could manipulate these two factors experimentally to investigate their effects on emotion word processing.

## 5. Conclusions

In summary, the present study examined the impact of AoA, valence, and arousal on emotion word recognition across two experiments. The results confirmed the effects of AoA, valence, and arousal on emotion word recognition. Importantly, AoA and valence exerted independent influences on emotion word recognition, with longer reaction times observed for later-acquired words and negative words compared to early-acquired words and positive words, respectively. In contrast, AoA and arousal interacted, with a larger AoA effect found for low-arousing words while the effect was insignificant for high-arousing words. These findings underscore the importance of considering AoA in emotion word processing and highlight the interplay between AoA and arousal. Moreover, the results suggest that the AoA effect may operate primarily at the perceptual level rather than the semantic level.

## Figures and Tables

**Table 1 behavsci-13-00568-t001:** Characteristics of critical stimuli for Experiment 1 (SD in brackets).

Group	Valence	AoA	Arousal	Strokes	wFreq (logCHR)
NE	−1.764 (SD: 0.316)	9.117 (SD: 0.951)	2.573 (SD: 0.469)	16.550 (SD: 4.242)	2.209 (SD: 0.376)
NL	−1.779 (SD: 0.319)	13.807 (SD: 0.578)	2.477 (SD: 0.475)	17.525 (SD: 3.700)	2.259 (SD: 0.426)
PE	1.754 (SD: 0.298)	9.073 (SD: 0.941)	2.550 (SD: 0.353)	17.500 (SD: 4.243)	2.259 (SD: 0.374)
PL	1.748 (SD: 0.306)	13.789 (SD: 0.813)	2.536 (SD: 0.398)	16.675 (SD: 4.009)	2.342 (SD: 0.426)

Note. NE: negative early-acquired word; NL: negative later-acquired word; PE: positive early-acquired word; PL: positive later-acquired word.

**Table 2 behavsci-13-00568-t002:** Descriptive statistics for Experiment 1 (SE in brackets).

	Early-Acquired		Later-Acquired	
	Positive	Negative	Positive	Negative
Reaction time	704 (15)	737 (18)	720 (15)	749 (16)
Accuracy rate	0.99 (0.01)	0.96 (0.01)	0.97 (0.01)	0.96 (0.01)

**Table 3 behavsci-13-00568-t003:** Characteristics of critical stimuli for Experiment 2 (SD in brackets).

Group	Arousal	AoA	Valence	Strokes	wFreq (logCHR)
HE	2.974 (SD: 0.279)	8.008 (SD: 1.074)	0.110 (SD: 1.155)	17.200 (SD: 4.040)	2.178 (SD: 0.484)
HL	2.973 (SD: 0.204)	14.310 (SD: 1.216)	0.286 (SD: 1.028)	17.975 (SD: 4.003)	2.148 (SD: 0.316)
LE	1.167 (SD: 0.140)	7.827 (SD: 1.165)	0.163 (SD: 0.281)	18.025 (SD: 5.245)	2.148 (SD: 0.330)
LL	1.190 (SD: 0.142)	14.220 (SD: 1.208)	0.216 (SD: 0.334)	16.775 (SD: 3.977)	2.050 (SD: 0.369)

Note: HE: high-arousing early-acquired words; HL: high-arousing later-acquired words; LE: low-arousing early-acquired words; LL: low-arousing later-acquired words.

**Table 4 behavsci-13-00568-t004:** Descriptive statistics for Experiment 2 (SE in brackets).

	Early-Acquired		Later-Acquired	
	High-Arousing	Low-Arousing	High-Arousing	Low-Arousing
Reaction time	646 (13)	665 (15)	652 (15)	708 (15)
Accuracy rate	0.98 (0.01)	0.99 (0.01)	0.98 (0.01)	0.94 (0.02)

## Data Availability

Data are available on request.

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
