# Peer review of "Beyond Valence and Arousal: The Role of Age of Acquisition in Emotion Word Recognition"

_behavsci, 2023, doi:10.3390/bs13070568_

Round 1

Reviewer 1 Report

The paper presents a study on the impact of the Age of Acquisition (AoA) factor on the recognition of emotional words. The authors conducted two experiments to examine the relationship between AoA and the valence/arousal of words in two tasks: valence judgment and discrimination of real versus fake words in high and low arousal categories. They argue that AoA did not interact with valence in the first experiment but did interact with arousal in the second experiment.

While the subject is worth investigating, there is a need for more detailed problem formulation.

One significant omission is the lack of a mathematical definition of "interaction." It would be beneficial to explicitly state the null hypothesis for this concept.

To enhance the comprehensibility of the paper, the following details of the experiments could be added:

- The specific sentence questions posed to the subjects.
- The range of arousal/valence values in the dataset and the units of measurement used (e.g., Table 1, line 225).
- Clarification on the meaning of "mean" used for filtering data (e.g., line 246 and line 328). Is it the mean of response time in seconds?
- Additionally, it is important to note that while the main concept of the paper is AoA, a clearer definition of AoA should be provided earlier in the text (when it stated in section 4.3 "early-acquired words around 9 years and later-acquired words around 13...").

Author Response

We would like to express our sincere gratitude for the invaluable feedback and constructive comments provided on our manuscript titled "Beyond Valence and Arousal: The role of Age of Acquisition on Emotion Word Recognition." We are truly appreciative of the time and effort invested by the reviewers in reviewing our work.

Based on their insightful suggestions, we have diligently revised the manuscript to address the raised concerns and improve its overall quality. Below, we present a detailed response to each of the reviewers' comments:

Reviewer 1

The paper presents a study on the impact of the Age of Acquisition (AoA) factor on the recognition of emotional words. The authors conducted two experiments to examine the relationship between AoA and the valence/arousal of words in two tasks: valence judgment and discrimination of real versus fake words in high and low arousal categories. They argue that AoA did not interact with valence in the first experiment but did interact with arousal in the second experiment.

While the subject is worth investigating, there is a need for more detailed problem formulation.

Thank you for your great advice. We have reformulated the problem in introduction (P. 4-5).

One significant omission is the lack of a mathematical definition of "interaction." It would be beneficial to explicitly state the null hypothesis for this concept.

Thank you for your kind reminder. We have added this information in introduction (P. 4).

To enhance the comprehensibility of the paper, the following details of the experiments could be added:

- The specific sentence questions posed to the subjects.

We have added the sentence with thanks (P. 6).

- The range of arousal/valence values in the dataset and the units of measurement used (e.g., Table 1, line 225).

Thank you for this suggestion. We have added the range in the text (P. 6).

- Clarification on the meaning of "mean" used for filtering data (e.g., line 246 and line 328). Is it the mean of response time in seconds?

Yes. We have added this information in the text (P. 6 and P. 8).

- Additionally, it is important to note that while the main concept of the paper is AoA, a clearer definition of AoA should be provided earlier in the text (when it stated in section 4.3 "early-acquired words around 9 years and later-acquired words around 13...").

Thank you for this insightful comment. We have introduced the definition of AoA in earlier in the text (P. 1).

Reviewer 3

The authors have submitted a concise article with results from two experiments. I find the paper to add important information regarding AoA and the processing of emotion words. Most of my comments are provided in comments on the PDF. Overall, a few major changes should be implemented, namely:

We are extremely grateful for your detailed comments throughout the manuscript. We have revised the whole manuscript accordingly (Please review all the highlighted text within the manuscript).

  1. the inclusion of relevant definitions and examples

Thank you for your great suggestion. We have provided related definitions and examples accordingly (P. 1-2).

  1. greater detail on methodology - this is very important for replicability

We have provided more details on methods with thanks (P. 5-6).

  1. greater detail on statistical design and inclusion of model outputs

We have provided more statistical design and outputs (P. 6-7). Thank you!

We firmly believe that the revisions made to the manuscript have substantially fortified its content, bringing it into closer alignment with the research objectives. We are confident that the adjustments we have implemented address the concerns raised by the reviewers and have resulted in a higher-quality publication. We would like to take this opportunity to express our sincere appreciation to the reviewers for their invaluable input, which undeniably contributed to the overall improvement of our work.

Yours sincerely

Authors

Reviewer 2 Report

Very clear study which deals with the influence of age of acquisition (AoA) and on how it modulates emotion word processing. Two experiments were conducted. The first one made use of a valence-judgment task to explore the roles of valence and AoA on emotion word processing. The second one employed  alexical decision task to analyze AoA as well as connected arousal effects. As a result, evidence is provided that AoA significantly influence emotion word processing. The interplay between AoA and arousal easily comes out.

The paper reaches the standards of robustness, quality, and novelty for MDPI Behavioral Sciences.

I recommend it for publication in the present form.

Author Response

We would like to express our sincere gratitude for the invaluable feedback and constructive comments provided on our manuscript titled "Beyond Valence and Arousal: The role of Age of Acquisition on Emotion Word Recognition." We are truly appreciative of the time and effort invested by the reviewers in reviewing our work.

We have revised the manuscript according to the quality check of the reviewer. 

We firmly believe that the revisions made to the manuscript have substantially fortified its content, bringing it into closer alignment with the research objectives. We are confident that the adjustments we have implemented address the concerns raised by the reviewers and have resulted in a higher-quality publication. We would like to take this opportunity to express our sincere appreciation to the reviewers for their invaluable input, which undeniably contributed to the overall improvement of our work.

Yours sincerely

Authors

Reviewer 3 Report

The authors have submitted a concise article with results from two experiments. I find the paper to add important information regarding AoA and the processing of emotion words. Most of my comments are provided in comments on the PDF. Overall, a few major changes should be implemented, namely:

1. the inclusion of relevant definitions and examples

2. greater detail on methodology - this is very important for replicability

3. greater detail on statistical design and inclusion of model outputs

Author Response

(The authors gave the same response as above.)
